# Low-Frequency Vibrational Spectroscopy Characteristic of Pharmaceutical Carbamazepine Co-Crystals with Nicotinamide and Saccharin

**DOI:** 10.3390/s22114053

**Published:** 2022-05-27

**Authors:** Meilan Ge, Yuye Wang, Junfeng Zhu, Bin Wu, Degang Xu, Jianquan Yao

**Affiliations:** 1Institute of Laser and Optoelectronics, School of Precision Instruments and Optoelectronics Engineering, Tianjin University, Tianjin 300072, China; meilange@tju.edu.cn (M.G.); xudegang@tju.edu.cn (D.X.); jqyao@tju.edu.cn (J.Y.); 2Key Laboratory of Optoelectronic Information Technology (Ministry of Education), Tianjin University, Tianjin 300072, China; 3Science and Technology on Electronic Test & Measurement Laboratory, Qingdao 266555, China; zhujfeng@mail.ustc.edu.cn (J.Z.); wubin@ei41.com (B.W.)

**Keywords:** low-frequency vibrational characteristic, low-wavenumber Raman spectroscopy, THz spectroscopy, pharmaceutical co-crystal

## Abstract

The pharmaceutical co-crystal has attracted increasing interest due to the improvement of physicochemical properties of active pharmaceutical ingredients. The characterization of pharmaceutical co-crystal is an integral part of the pharmaceutical field. In this paper, the low-frequency vibrational properties for carbamazepine co-crystals with nicotinamide and saccharin (CBZ-NIC and CBZ-SAC) have been characterized by combining the THz spectroscopy with low-wavenumber Raman spectroscopy. The experiment results show that, compared with the individual constituents, CBZ-NIC and CBZ-SAC co-crystals not only have different characteristic absorption peaks in the 0.3-2.5 THz region, but also have significant low-wavenumber Raman characteristic peaks in 0–100 cm^−1^. Density functional theory was performed to simulate the terahertz and low-wavenumber Raman spectra of the two co-crystals, where the calculation agreed well with the measured vibrational peak positions. The vibrational modes of CBZ-NIC and CBZ-SAC co-crystals were assigned through comparing theoretical results with the experimental spectra. Meanwhile, the low-frequency infrared and/or Raman active of characteristic peaks for such co-crystals were discussed. The results indicate the combination of THz spectroscopy and low-wavenumber Raman spectroscopy can provide more comprehensive low-frequency vibrational information for pharmaceutical co-crystals, such as collective vibration and skeleton vibration, which could play an important role in pharmaceutical science.

## 1. Introduction

Carbamazepine (CBZ) is a common psychotropic drug, which is preferred in the treatment of epilepsy and has had better therapeutic effects in treating trigeminal neuralgia and bipolar affective disorder. As a typical oral drug, however, CBZ shows low solubility and limited bioavailability [1,2]. The pharmaceutical co-crystal is an effective approach to improve CBZ solubility and bioavailability, which form the combination between active pharmaceutical ingredients (APIs) and co-crystal conformers (CCFs) by hydrogen bonding or intermolecular interactions [3,4,5,6]. CBZ co-crystals show unique physicochemical properties without altering the chemical nature and bioactivity of CBZ, such as high solubility and bioavailability [7,8,9,10]. In the past, a series of co-crystals between CBZ and various CCFs have been reported [11,12].

Currently, the basic physicochemical properties of pharmaceutical co-crystals are usually characterized by powder X-ray diffraction (PXRD), differential scanning calorimetry (DSC), infrared spectroscopy (IR) and Raman spectroscopy. PXRD, as a common method for verifying the formation of co-crystals, can directly obtain structure information about atoms or molecules inside CBZ co-crystals, but it has no chemistry information [13]. The DSC method can acquire a range of thermodynamic data, but sample is destroyed during analysis [14]. Considering that IR and Raman spectroscopy can probe energy levels of molecules associated with chemical bonds and provide fingerprint characteristic of materials [7], they have been widely applied in characterizing vibration and/or rotation modes of CBZ co-crystals due to their high sensitivity, label-free and non-destructive characteristics [15]. Currently, the intra-molecular vibrations of the functional groups of CBZ co-crystals ranging from 200 cm^−1^ to 1800 cm^−1^ have been characterized by IR and traditional Raman spectroscopy techniques. Comparatively, low-frequency vibrational information plays an essential role in revealing intra-molecular and inter-molecular weak interactions, which involve the configuration and conformation of molecules associated with biological activity [16]. Particularly, the low-frequency information can identify the differences between isomers, which arises from the hydron atom in molecular structure [17]. Therefore, the study of low-frequency vibrational information based on spectroscopy technique is of importance to biomedicine. However, there are few studies on the low-frequency information in the 0–200 cm^−1^ region.

Terahertz (THz) spectroscopy and low-frequency Raman spectroscopy are appealing methods for the study of low-frequency information due to their nondestructive and label-free nature. THz spectroscopy can reflect the information of the absorption of the THz wave by the asymmetrical vibration and/or rotation energy level, the polar group vibration and/or the rotation energy level of molecules. Thus, THz spectroscopy is sensitive to the weak intermolecular interactions, environment information and collective vibration in the molecular level within the samples [6,18]. Although THz spectroscopy has limited resolution (~7.6 GHz ≈ 25.3 cm^−1^), it has been used to characterize CBZ co-crystals [5]. It is well known that the analysis of the assignments of THz characteristic peaks is the major means to study the vibration and/or rotation information of co-crystals themselves. However, the specific assignments of THz characteristic peaks still remain unclear. Low-wavenumber Raman spectroscopy can provide complimentary vibration information in a similar spectral range, which can characterize scattering of excitation light caused by the symmetrical vibration and/or rotation energy level, nonpolar groups vibration and /or energy level of molecules. Considering that low-wavenumber Raman spectroscopy is an efficient tool for the study of collective translation and deformation of the molecular skeleton inside the crystal lattice or deformation of the whole unit cell [19,20,21], it is potentially an effective method for characterizing the low-frequency vibrational characteristics of CBZ co-crystals. In particular, the resolution of low-frequency Raman spectrum can reach 2.5 cm^−1^, which can obtain more abundant low-frequency vibrational information. Therefore, comprehensive THz and Raman spectroscopy of CBZ co-crystals can obtain richer low-frequency vibrational information.

In this study, the low-frequency properties of CBZ co-crystals with nicotinamide (NIC) and saccharin (SAC) were characterized by combining the THz time-domain spectroscopy (THz-TDS) and low-wavenumber Raman spectroscopy. Compared with the spectra of the individual constituents of CBZ, NIC and SAC, the CBZ-NIC and CBZ-SAC co-crystals have significant absorption characteristic peaks in the 0.3–2.5 THz region and Raman characteristic peaks in the 0–100 cm^−1^ region. Density functional theory (DFT) was used to simulate the terahertz and low-wavenumber Raman spectra of the two co-crystals, and then the assignment of vibrational peaks was performed. Moreover, the low-frequency infrared and/or Raman active of characteristic peaks for such co-crystals were discussed. These results show that a combination of the THz spectroscopy and low-wavenumber Raman spectroscopy can obtain comprehensive absorption and scattering low-frequency vibrational information of CBZ co-crystals, such as the collective vibration and skeleton vibration, which has important significance for the further investigation of pharmaceutical science and other fields.

## 2. Materials and Methods

### 2.1. Materials

Carbamazepine (98%), nicotinamide (99%), saccharin (99%) and polyethylene (grain size 40–48 μm) were purchased from Sigma-Aldrich (Shanghai, China). The solvent was of analytical grade. All compounds were used as received without further purification.

CBZ-NIC and CBZ-SAC co-crystals were prepared by the slow solvent evaporation method according to a previously reported method [22], as shown in Figure 1a. Equimolar APIs (CBZ, 236 mg, 1 mmol) and CFFs (NIC, 122 mg, 1 mmol or SAC, 183 mg, 1 mmol) were dissolved in an amount of ethanol (40 mL) and stirred for 3 h. The solution was slowly evaporated at room temperature and then co-crystals were obtained after several days. As for the Raman spectroscopy measurement, the powder samples were directly used without any treatment. For THz spectroscopy measurement, the co-crystals tablet mixed with white polyethylene was prepared with a mixing ratio of 1:1. As shown in Figure 1b, the mixture was grinded and then filtrated through a 200-mesh sieve to minimize the scattering effects from sample particles during spectral measurements. The mixed powder was pressed into circular tablets around 10 mm diameter and 1.0 mm thickness under 6 MPa pressure for 3 min. Five replicates of each sample were prepared for reproducibility assessment.

### 2.2. Apparatus and Procedure

In order to confirm the formation of the co-crystal, CBZ-NIC and CBZ-SAC co-crystals were identified using PXRD. PXRD patterns of the two co-crystals were recorded with the Rigaku Smartlab 9 KW diffraction system using a Cu Kα source (λ = 0.15406 nm) over the 2θ range from 20° to 70°. The scanning speed was set to 3°/s.

Low-wavenumber Raman spectra were measured using a commercial THz-Raman microscope system, as shown in Figure 1c. This system mainly contains an integrated laser module compatible (Ondax, Monrovia, CA, USA), a microscope (Leica, Wetzlar, Germany) and a spectrometer (Horiba, Japan) equipped with a CCD detector cooled to −50 ℃ (Syncerity, Horiba, Kyoto, Japan). The integrated laser module compatible comprises of a stable 785 nm laser as an excitation light source and a series of high-throughput volume holographic grating (VHG) filters. Briefly, two VHG amplified spontaneous emission (ASE) filters (NoiseBlockTM, Ondax, Inc., Monrovia, CA, USA) were used to remove ASE from the laser. The beam was focused on the samples by a 10× objective lens. Then the 180° backscattered light from sample was collected and filtered through the VHG beam splitter towards the two ultra-narrowband VHG notch filters (SureLockTM, Ondax, Inc.), which can further remove the collected Ralyeigh scattered light [23]. The filtered signal was focused into a spectrometer within a 1200 grooves/mm grating via a fiber. The Raman scattered light was detected by a CCD detector. The measurement range was from −200 cm^−1^ to 2000 cm^−1^ with a spectral resolution of 2.5 cm^−1^. Each Raman spectrum was recorded using 70 mW laser power with 10 s acquisition times. In order to compare the Raman spectrum with the THz spectrum, the analysis range of the Raman spectrum was chosen as 0–100 cm^−1^. The final Raman spectrum of each sample was the averaged result of five measurements, where the spectrum of each measurement was acquired at different positions. The experimental temperature was maintained at room temperature during the whole experiment.

The THz spectrum was obtained using the THz time-domain spectrometer (Advantest Corp., TAS7500SP, Tokyo, Japan), as shown in Figure 1d. A pair of GaAs photoconductive antennas as the THz emitter and detector were driven by a femtosecond laser with the center wavelength of 800 nm. The emitted THz radiation was collimated and focused on samples by parabolic mirrors. Then, the signal transmitted through the sample was collected by THz detector. The spectrum was measured from 0.3 to 2.5 THz with a frequency resolution of 7.6 GHz and dynamic range of 70 dB. All the experimental results were obtained using transmission mode at room temperatures. The relative humidity was always kept below 3% by purging the drying air during the measurements in order to prevent the effect of the absorption of atmospheric vapor. The final spectrum of each sample was the average of five measurements. The time domain signals of each sample and reference (without sample) were converted to frequency domain signals through the Fourier transform. Then, the THz spectrum was the result of the sample frequency signal divided by corresponded reference [24].

### 2.3. Theoretical Calculation

Quantum chemical computation of theoretical structures was carried out using the DFT. The initial structures of the CBZ-NIC and CBZ-SAC co-crystals were obtained from the Cambridge Crystallographic Data Centre (CCDC), as shown in Figure 2. The CCDC number of CBZ-NIC and CBZ-SAC are 1544195 and 1562073, respectively. The simulated low-wavenumber Raman and THz spectra of the CBZ-NIC and CBZ-SAC co-crystals were calculated with the B3LYP functional and 6-311++G (d, p) basis set. Lorentzian line shapes were convolved into the calculated vibrational modes using a full-width half-maximum (FWHM) value of 4.0 cm^−1^ [25,26].

## 3. Results and Discussion

### 3.1. PXRD Analysis

The PXRD patterns for the CBZ-NIC and CBZ-SAC co-crystals with their individual constituents were shown in Figure 3. It was obvious that there were significant characteristic diffraction peaks in the 10–35° range. The PXRD patterns of CBZ-NIC and CBZ-SAC co-crystals with their individual constituents were consistent with the literature, which verified the reliability of our prepared co-crystals [3,5].

### 3.2. THz Absorption Spectral Characteristic and Analysis of CBZ-NIC and CBZ-SAC Co-Crystals

Figure 4 showed THz spectra of CBZ-NIC and CBZ-SAC co-crystals with their individual constituents in the 0.3–2.5 THz region, where the positions of all characteristic peaks were labeled. It can be seen from Figure 4a that CBZ as an API has three characteristic peaks at 1.23, 1.82 and 2.04 THz, in which the peaks at 1.23 and 1.82 THz show strong THz absorption characteristic, while the peak at 2.04 THz is relative weak. The NIC has one very narrow strong THz absorption peak at 1.06 THz, one strong peak at 1.93 THz and three weak peaks at 0.60, 1.70 and 2.39 THz, respectively. The THz spectra of CBZ and NIC were in good agreement with the previous work except an obvious characteristic peak of NIC at 2.39 THz [5]. The SAC exhibits three weak peaks at 1.13, 1.89 and 2.01 THz, as shown in Figure 4b. However, the measured spectral properties at 1.89 and 2.01 THz were different from that in Ref [5], where a broadband characteristic peak at 1.98 THz appeared.

THz spectra of co-crystals exhibit different characteristics from their individual constituents. From Figure 4a, the CBZ-NIC co-crystal has two strong THz peaks at 1.56 THz and 1.91 THz, one relatively weak peak at 1.09 THz, and three weak peaks at 0.41, 1.63 and 2.26 THz. Comparatively, the CBZ-SAC co-crystal shows five peaks at 1.04, 1.46, 1.68, 2.22 and 2.35 THz, in which the peaks at 1.04, 1.68 and 2.35 THz exhibit strong THz absorption characteristics and two weak peaks appear at 1.46 and 2.22 THz, as shown in Figure 4b. It is obvious that those absorption features of the co-crystal do not derive from the superposition of THz spectra of their individual constituents. It could be inferred that the inter-molecular interactions between the CBZ and CCFs result in the structure change of the unit cells between the raw materials, which can be characterized by THz spectroscopy [26]. In addition, it should be mentioned that the new THz characteristic peaks of CBZ-NIC and CBZ-SAC co-crystals were detected compared with the results in the reference [5]. It may be due to the difference of spectral resolution and measurement range for experimental systems.

The DFT theoretical spectra of CBZ-NIC and CBZ-SAC co-crystals are shown in Figure 5. For better comparison, the THz spectra of CBZ-NIC and CBZ-SAC in Figure 4 were depicted again in Figure 5. At the same time, all of these vibrational mode distributions are presented in Table 1 and Table 2. As shown in Figure 5a, the CBZ-NIC co-crystal has five THz absorption peaks at the positions of 0.32, 0.82, 1.62, 2.07 and 2.56 THz in the theoretical spectrum, which are essentially in agreement with the experimental result.

The experimental peak at 0.41 THz corresponds to the vibrational mode calculated at 0.32 THz, which arises from CBZ and NIC molecules’ collective twisting vibration. The experimental peak at 0.63 THz can be attributed to the simulated mode at the position at 0.82 THz, arising from the collective out-of-plane rocking vibration of the CBZ and NIC molecules. The characteristic peak at 1.56 THz corresponds to the theoretical spectrum at 1.62 THz, which is due to the strong collective out-of-plane rocking vibration of CBZ and the weak collective in-plane rocking vibration of NIC. The experimental spectral feature at 1.91 THz can be attributed to the calculated vibration mode at 2.07 THz, which is caused by the NIC molecule’s strong collective twisting vibration and the CBZ molecule’s weak shearing vibration. The peak at the position 2.26 THz in the experimental result corresponds to the calculated mode at 2.56 THz, arising from the strong out-of-plane rocking vibration of N2-6C=10O and the twisting of C7-16C-17C-19C-21C-30C within CBZ molecules, and the weak collective in-plane rocking vibration of the NIC molecule.

As shown in Figure 5b, the CBZ-SAC co-crystal also has five THz absorption characteristic peaks at the positions of 0.23, 0.86, 1.65, 2.00 and 2.50 THz in the theoretical spectrum. The experimental THz peak at 1.04 THz can be attributed to the calculated vibration mode at 0.86 THz, which is due to the collective in-plane shearing vibration of the CBZ and SAC molecules. The characteristic peak at 1.68 THz corresponds to the calculated spectrum at 1.65 THz, which is caused by the strong bending vibration of CBZ, the in-plane rocking vibration of O2=S1=O3 within the SAC, and the weak out-of-plane rocking vibration of SAC. The experimental peak at 2.22 THz corresponds to the theoretical spectrum at 2.00 THz, which is caused by the bending vibration of C45-C36-C34-C32-C31-C22 belonging to the CBZ molecule. The experimental spectral feature at 2.35 THz corresponds to the simulated peak at 2.50 THz, arising from the out-of-plane rocking vibration of N19-21C=O18 and the twisting vibration of C45-C36-C34-C32-C31-C22 of the CBZ molecule. It is noted that, comparing the experimental and theoretical spectra of the CBZ-NIC and CBZ-SAC co-crystals, the relative intensities of absorption peaks are different, and the THz characteristic peaks shifted a little. This can be attributed to the fact that the theoretical calculation was performed at an absolute 0 degree temperature, whereas the experimental spectrum was obtained under room temperature [27]. It is demonstrated that THz spectroscopy is sensitive to the collective molecular vibration, weak molecular vibration and skeleton vibration.

Additionally, the absorption peak of the CBZ-NIC co-crystal at 1.09 THz and the absorption peak of the CBZ-SAC co-crystal at 1.46 THz in the measured spectra does not appear in the corresponding theoretical spectra. It might be that the theoretical calculation is only of a single molecule unit, and ignores intermolecular forces within solid-state crystalline unit cells [27].

### 3.3. Low-Wavenumber Raman Spectral Characteristic and Analysis of CBZ-NIC and CBZ-SAC Co-Crystals

Figure 6 shows the experimental low-wavenumber Raman spectra of CBZ-NIC and CBZ-SAC co-crystals with their individual constituents, where the position of all characteristic peaks were labeled. It is obvious that the CBZ molecule has five Raman characteristic peaks in the 0–100 cm^−1^ region, which contains one strong peak at 39.6 and 90.4 cm^−1^, and three relatively weak peaks at 47.9, 65.3 and 74.0 cm^−1^. The low-wavenumber Raman spectrum of NIC has three characteristic peaks at 26.1, 50.5 and 74.8 cm^−1^, in which the peak at 26.1 cm^−1^ exhibits strong and sharp characteristic, the peak at 74.8 cm^−1^ is relative weak and the peak at 50.5 cm^−1^ is very weak, as shown in Figure 6a. From Figure 6b, SAC shows three strong characteristic peaks at 24.3, 50.1 and 56.9 cm^−1^, a strong shoulder peak at 64.7 cm^−1^, and two weak peaks at 79.4 and 90.7 cm^−1^. For CBZ-NIC and CBZ-SAC co-crystals, they exhibit different Raman features from their individual constituents, as shown in Figure 6a,b, respectively.

The CBZ-NIC co-crystal exhibits three strong Raman characteristic peaks at 23.4, 43.6 and 73.0 cm^−1^, and two weak intensity peaks at 33.5 and 51.9 cm^−1^ in the 0–100 cm^−1^ range. The CBZ-SAC co-crystal has six peaks, which consists of three strong peaks at 29.2, 50.8 and 76.1 cm^-1^, one peak at 64.6 cm^−1^ and a peak at 41.1 cm^−1^ with a shoulder at 37.1 cm^−1^. The results indicate that the CBZ co-crystal is a new compound with hydrogen bonds, π-π stacking and other inter-molecular interaction between the starting parent materials. Additionally, the CBZ co-crystal leads to the changes of the molecular structures of CBZ and CCFs, which can be detected by the low-wavenumber Raman spectroscopy.

A comparison of the theoretical DFT calculation and the experimental Raman spectral results are shown in Figure 7. The characteristic vibrational peaks of the CBZ-NIC and CBZ-SAC co-crystals shown in the low-wavenumber Raman spectra are summarized in detail in Table 3 and Table 4, respectively. The CBZ-NIC co-crystal has five low-wavenumber Raman characteristic peaks at the position of 9.2, 27.4, 53.7, 73.8 and 91.3 cm^−1^ in the theoretical spectrum, as shown in Figure 7a. The peak at the position 23.4 cm^−1^ in the experimental result corresponds to the peak at 27.4 cm^−1^ in the theoretical spectrum, which is caused by the collective out-of-plane rocking vibration of CBZ and NIC molecules.

The experimental peak at 51.9 cm^−1^ can be attributed to the simulated mode at the position 53.7 cm^−1^, arising from strong collective out-of-plane rocking vibration of CBZ molecules, and weak collective in-plane rocking vibration of NIC. The experimental peak at 73.0 cm^−1^ corresponds to the calculated the peak at 73.8 cm^−1^. This peak is due to the bending vibration of C7-16C-17C-19C-21C-30C within CBZ molecules, and weak in-plane rocking vibration of NIC molecules.

CBZ-SAC co-crystal has seven low-wavenumber Raman characteristic peaks at the positions of 8.7, 21.5, 28.6, 54.2, 65.5, 91.1 and 99.6 cm^−1^ in the theoretical spectrum, as shown in Figure 7b. The characteristic peak of the CBZ-SAC co-crystal at 29.2 cm^−1^ in the experimental result corresponds to the peaks calculated at 28.6 cm^−1^, which is due to the collective in-plane shearing vibration of CBZ and SAC molecules. The experimental spectral feature at 50.8 cm^−1^ can attributed to the simulated result at 54.2 cm^−1^, which arises from a combination of CBZ molecules’ strong bending vibration and the in-plane rocking vibration of O2=S1=O3 of SAC molecules. The experimental Raman peak at 64.6 cm^−1^ corresponded to the calculated vibrational mode at 65.5 cm^−1^ arises from the bending vibration of C45-C36-C34-C32-C31-C22, which belongs to CBZ molecules. These results show that the low-wavenumber Raman spectroscopy can characterize the collective vibration of molecules of different types, weak molecular vibration and molecular skeleton vibration with low-frequency Raman characteristic.

It should be noted that there is still a significant gap between the experimental and the calculated results of the two co-crystals. The CBZ-NIC co-crystal characteristic peaks at 9.2 and 91.3 cm^−1^ and the CBZ-SAC co-crystal peaks at 8.7, 21.5, 91.1 and 99.6 cm^−1^ in the theoretical spectra do not appear in the corresponding experimental spectra. Furthermore, the CBZ-NIC co-crystal peak at 43.6 cm^−1^ and the CBZ-SAC co-crystal peaks at 37.1, 41.1 and 76.1 cm^−1^ in the experimental spectra do not match with the corresponding calculated result spectra. It may be deduced from two reasons. On one hand, the theoretical calculated results were obtained under the condition of 0 K, whereas the experiment results were obtained under the room temperature. On the other hand, the calculation modes used here were based on the single-molecule structure, and inter-molecular forces within solid-state crystalline unit cell were ignored. Therefore, the structure model should be optimized in future work.

### 3.4. Comparison of THz and Low-Wavenumber Raman Characteristic of CBZ-NIC and CBZ-SAC Co-Crystals

In order to further understand the low-frequency characteristic, we compared the THz absorption spectra and low-wavenumber Raman spectra of CBZ-NIC and CBZ-SAC co-crystals, respectively, as shown in Figure 8. For the CBZ-NIC co-crystal, the THz absorption peak at 0.63 and 1.56 THz coincided with the Raman peaks at 23.4 cm^−1^ (0.70 THz) and 51.9 cm^−1^ (1.56 THz), respectively, as shown in Figure 8a. For the CBZ-SAC co-crystal, the THz absorption peaks at 1.04 THz, 1.46 THz and 2.22 THz correspond to the Raman peaks at 29.2 cm^−1^ (0.88 THz), 50.8 cm^−1^ (1.53 THz) and 64.6 cm^−1^ (1.94 THz), respectively, as shown in Figure 8b. (The red line in the figure shows the peaks that can be matched). Here, a little difference between THz frequency and Raman mode frequency is allowed owing to the influence of factor group splitting [28,29].

Comparing Table 1 with Table 3 for CBZ-NIC co-crystal, it is seen that Raman characteristic peak at 23.4 cm^−1^ and THz absorption peak at 0.63 THz both arise from the collective out-of-plane rocking vibration of CBZ and NIC molecules; the Raman peak at 51.9 cm^−1^ and THz absorption peak at 1.56 THz both arise from a combination of the strong collective out-of-plane rocking vibration of CBZ molecules and the weak in-plane collective rocking vibration of NIC molecules. These demonstrate that the two characteristic peaks of CBZ-NIC co-crystals are not only infrared active, but also Raman active. Furthermore, comparing Table 2 with Table 4 for CBZ-SAC co-crystals, it is obvious that the characteristic peak at 1.04 THz of the THz spectrum and the peak at 29.2 cm^−1^ of the Raman spectrum are both attributed to the collective in-plane shearing vibration of CBZ and SAC molecules. The Raman peak at 50.8 cm^−1^ and THz absorption peak at 1.68 THz both come from the strong bending vibration within CBZ molecules, in-plane rocking vibration of O2=S1=O3 of SAC molecules, and the weak out-of-plane rocking vibration of SAC molecules. The Raman characteristic peak at 64.6 cm^−1^ and the THz absorption peak at 2.22 THz both derive from the bending vibration of C45-C36-C34-C32-C31-C22 that belongs to CBZ molecules. Thus, it is concluded that the three characteristic peaks of CB-SAC co-crystal are both infrared and Raman active.

Moreover, it is seen from Figure 8a,b that the THz absorption peaks of CBZ-NIC co-crystal at 0.41 THz, 1.09 THz, 1.91 THz and 2.26 THz and the THz absorption peaks of CBZ-SAC co-crystal at 1.46 THz and 2.35 THz did not appear in their corresponded Raman spectrum, which had different assignments of vibration modes. It indicates that these THz characteristic peaks of CBZ-NIC and CBZ-SAC co-crystals only are infrared active. Similarly, the Raman peaks at 33.5cm^−1^, 43.6 cm^−1^ and 73.0 cm^−1^ of CBZ-NIC co-crystal and the Raman peaks at 37.1 cm^−1^, 41.1 cm^−1^ and 76.1 cm^−1^ of CBZ-SAC co-crystal only are Raman active but not infrared active, where no corresponding peak appeared in the THz spectra. It can be demonstrated that THz absorption spectrum and low-wavenumber Raman spectrum are complementary to each other in characterizing the low-frequency characteristic of co-crystal. Therefore, a combination of the THz spectroscopy and low-wavenumber Raman spectroscopy is able to characterize more comprehensive low-frequency information both with Raman active and infrared active characteristics, such as collective vibration, weak molecular vibration and skeleton vibration of co-crystals.

Particularly, the determination of the infrared and Raman peaks of co-crystal not only depends on the characteristic peak position of experimental spectra, but also the theoretical vibrational modes assignment. For the CBZ-NIC co-crystal, although the position of the THz absorption peaks at 1.09 THz and 2.26 THz are close to the Raman characteristic peak at 33.5 cm^−1^ and 73.0 cm^−1^, respectively, they are not both infrared and Raman active. Moreover, the THz absorption peak at 2.35 THz of the CBZ-SAC co-crystal does not coincide with the Raman peak at 76.1 cm^−1^, though their positions are close. It is because the vibrational modes assignment of the THz absorption spectra is different from the low-wavenumber Raman peaks of the co-crystal.

## 4. Conclusions

In summary, the low-frequency vibrational characteristics of two CBZ co-crystals have been demonstrated by combining THz absorption spectroscopy and low-wavenumber Raman spectroscopy. The experimental results show that CBZ-NIC and CBZ-SAC co-crystals with their individual constituents have significant THz and low-wavenumber Raman characteristic peaks in the 0.3–2.5 THz and 0–100 cm^−1^ regions, respectively, which provide fingerprint spectral information of co-crystal and starting parent materials. The vibrational mode assignment of THz and Raman peaks has been analyzed using DFT calculation. The simulation shows a better agreement with the measured vibrational peak positions. Especially, the low-frequency infrared and/or Raman active of the characteristic peaks for CBZ-NIC and CBZ-SAC co-crystals were given based on the experimental and theoretical results. It could be indicated that the combination of THz and low-wavenumber Raman spectroscopy can characterize richer low-frequency vibrational properties, such as molecular collective vibration and skeleton vibration. These results provide theoretical and experimental benchmarks for vibrational spectroscopic studies in many fields. For example, this technique can be used in the real-time reaction monitoring of crystal form, phase or structural formations during the formulation of chemicals and polymers.

## Figures and Tables

**Figure 1 sensors-22-04053-f001:**
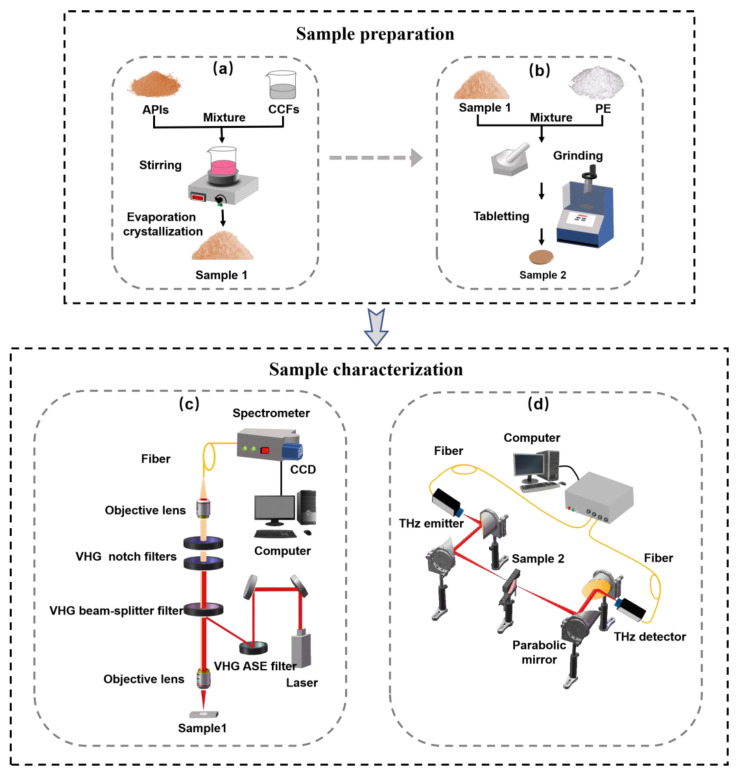
The schematic of the sample preparation and experimental system. (**a**) Co-crystals preparation process; (**b**) tableting process; (**c**) THz-Raman microscope system; (**d**) THz time-domain spectroscopy system.

**Figure 2 sensors-22-04053-f002:**
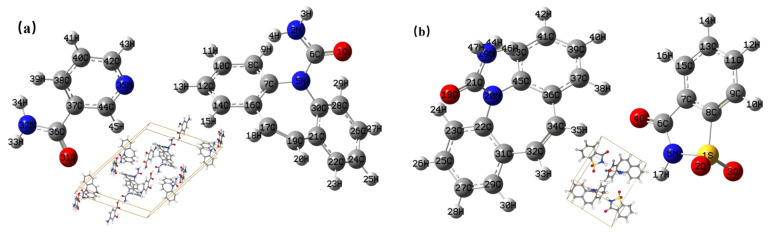
Molecular structures of the co-crystals. (**a**) CBZ and NIC; (**b**) CBZ and SAC.

**Figure 3 sensors-22-04053-f003:**
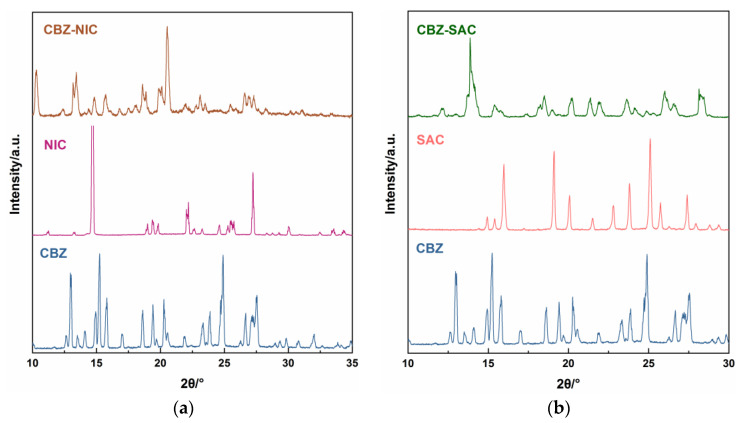
Experimental PXRD patterns of (**a**) CBZ-NIC and (**b**) CBZ-SAC co-crystals with their individual constituents.

**Figure 4 sensors-22-04053-f004:**
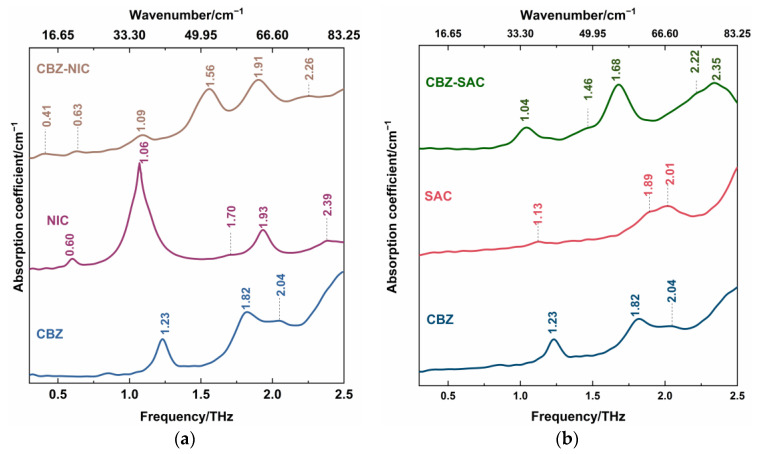
THz spectra of (**a**) CBZ-NIC and (**b**) CBZ-SAC co-crystals with their individual constituents.

**Figure 5 sensors-22-04053-f005:**
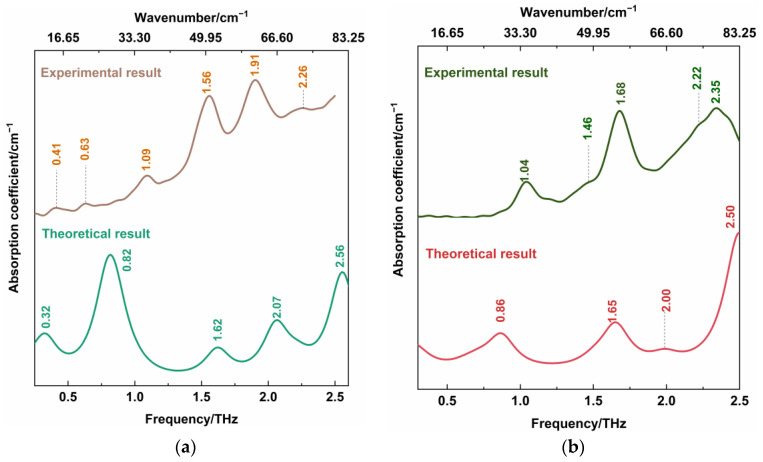
Comparison of THz spectra between theoretical and experimental results of the (**a**) CBZ-NIC co-crystal and (**b**) CBZ-SAC co-crystals.

**Figure 6 sensors-22-04053-f006:**
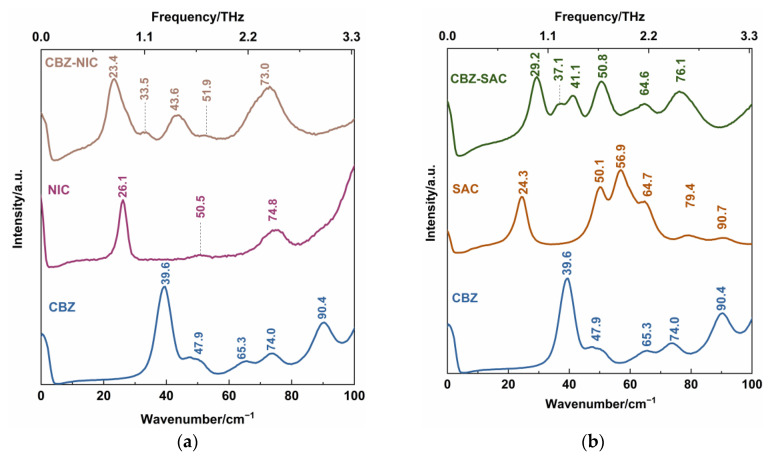
Low-wavenumber Raman spectra of (**a**) CBZ-NIC and (**b**) CBZ-SAC co-crystals with their individual constituents.

**Figure 7 sensors-22-04053-f007:**
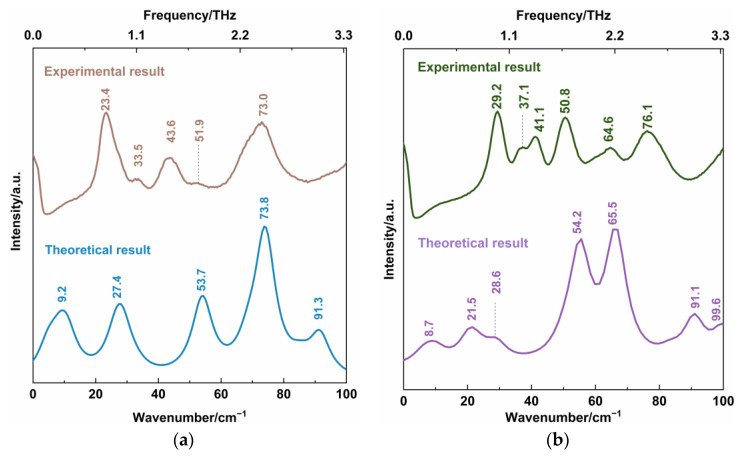
Comparison of low-wavenumber Raman spectra between theoretical and experimental results of (**a**) CBZ-NIC co-crystal and (**b**) CBZ-SAC co-crystals.

**Figure 8 sensors-22-04053-f008:**
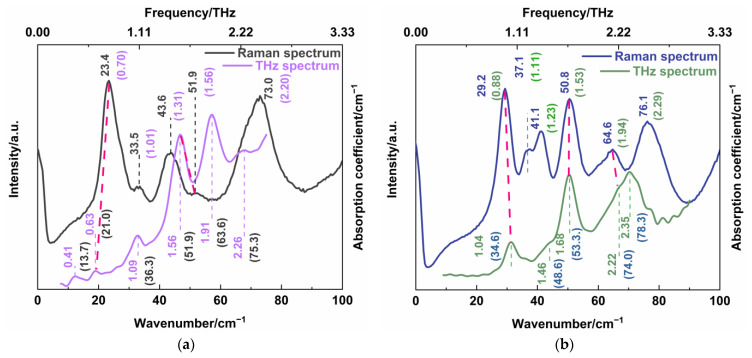
Comparison of THz absorption spectra and low-wavenumber Raman spectra of (**a**) CBZ-NIC and (**b**) CBZ-SAC co-crystals.

**Table 1 sensors-22-04053-t001:** Vibrational mode assignment of the CBZ-NIC co-crystal show in THz spectrum.

Experimental Result/THz	Calculation Result/THz	Vibrational Mode Assignment
0.41	0.32	CBZ and NIC molecule collective twisting vibration
0.63	0.82	collective out-of-plane rocking vibration of the CBZ and NIC molecules
1.09	—	—
1.56	1.62	strong collective out-of-plane rocking vibration of CBZ; weak collective in-plane rocking vibration of NIC
1.91	2.07	strong NIC molecule collective twisting vibration; weak CBZ molecule shearing vibration
2.26	2.56	strong out-of-plane rocking vibration of N2-6C=10O and twisting of C7-16C-17C-19C-21C-30C within CBZ molecules, and weak collective in-plane rocking vibration of NIC molecules

**Table 2 sensors-22-04053-t002:** Vibrational mode assignment of the CBZ-SAC co-crystal shown in THz spectrum.

Experimental Result/THz	Calculation Result/THz	Vibrational Mode Assignment
1.04	0.86	collective in-plane shearing vibration of CBZ and SAC molecules
1.46	—	—
1.68	1.65	strong bending vibration of CBZ; in-plane rocking vibration of O2=S1=O3 within the SAC; weak out-of-plane rocking vibration of SAC
2.22	2.00	bending vibration of C45–C36–C34–C32–C31–C22 belonging to CBZ molecules
2.35	2.50	out-of-plane rocking vibration of N19–21C=O18 and twisting vibration of C45–C36–C34–C32–C31–C22 of CBZ molecules

**Table 3 sensors-22-04053-t003:** Vibrational mode assignment for low-wavenumber Raman characteristic peaks of the CBZ-NIC co-crystal.

Experimental Result/cm^−1^	Calculation Result/cm^−1^	Vibrational Mode Assignment
	9.2	collective bending vibration of CBZ and NIC molecules
23.4	27.4	collective out-of-plane rocking vibration of CBZ and NIC molecules
43.6		
51.9	53.7	strong collective out-of-plane rocking vibration of CBZ molecules; weak collective in-plane rocking vibration of NIC
73.0	73.8	bending vibration of C7-16C-17C-19C-21C-30C within CBZ molecules; weak in-plane rocking vibration of NIC molecules
	91.3	twisting vibration of C7-16C-17C-19C-21C-30C belonging to CBZ molecules; weak in-plane rocking vibration of NIC molecules

**Table 4 sensors-22-04053-t004:** Vibrational mode assignment for low-wavenumber Raman characteristic peaks of the CBZ-SAC co-crystal.

Experimental Result/cm^−1^	Calculation Result/cm^−1^	Vibrational Mode Assignment
	8.7	CBZ and SAC molecules’ collective shearing vibration
	21.5	collective out-of-plane rocking vibration of CBZ and SAC molecules
29.2	28.6	collective in-plane shearing vibration of CBZ and SAC molecules
37.1		
41.1		
50.8	54.2	CBZ molecules’ strong bending vibration; in-plane rocking vibration of O2=S1=O3 of SAC molecules; weak out-of-plane rocking vibration of SAC molecules
64.6	65.5	bending vibration of C45-C36-C34-C32-C31-C22 which belongs to CBZ molecules
76.1		
	91.1	out-of-plane rocking vibration of N19–21C=O18 twisting vibration of C45–C36–C34–C32–C31–C22 belonging to CBZ molecules
	99.6	out-of-plane rocking vibration of O2=S1=C8 belonging to SAC molecules

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
