# Peer review of "Low-Frequency Vibrational Spectroscopy Characteristic of Pharmaceutical Carbamazepine Co-Crystals with Nicotinamide and Saccharin"

_sensors, 2022, doi:10.3390/s22114053_

Round 1

Reviewer 1 Report

The  manuscript  Low-frequency Vibrational Spectroscopy Characteristic of Pharmaceutical Carbamazepine Co-crystals with Nicotinamide and Saccharin by  Yuye Wang et al. focuses  on the low –frequency  THz and Raman  characteristics  of the  co-crystal forms. Correctly the authors have  used  PXRD prior to spectroscopic analyses. The manuscript is  overall  well  written and  may be considered for  publication after  minor corrections .

The authors claim to have used  the  crystal  structure for calculation  while  later on a  single molecule calculation is  described, Why not use the  PBC for better approximation ?   

Would  argue  about the  Raman calculations been  correct, they do not match the  experimental  ones. Especially in the case of  CNZ-SAC experimental has more  bands. Thus  the  assignment of the  Raman characteristic peaks should be revisited. It will be probably very nice  to verify the Raman DFT based on the comparison with the NIC , SAC and CNZ  only.

Minor

L64, L73 Cm-1 - -1 must be superscript etc …

Figure  3 For comparison  figure 3 must include not just the PXRD of the  co-crystals but also  the  ones  generated  from the CCDC single crystal structure .  The  PXRD of the starting  APIs may be provided as Supporting  information. Something similar has been  provided  for THz ( figure 4).

There are no conclusions  but Discussion?

Thats all 

Author Response

The  manuscript  Low-frequency Vibrational Spectroscopy Characteristic of Pharmaceutical Carbamazepine Co-crystals with Nicotinamide and Saccharin by  Yuye Wang et al. focuses  on the low –frequency  THz and Raman  characteristics  of the  co-crystal forms. Correctly the authors have  used  PXRD prior to spectroscopic analyses. The manuscript is  overall  well  written and  may be considered for  publication after  minor corrections .

Point 1: The authors claim to have used  the  crystal  structure for calculation  while  later on a  single molecule calculation is  described, Why not use the  PBC for better approximation ?   

Would  argue  about the  Raman calculations been  correct, they do not match the  experimental  ones. Especially in the case of  CNZ-SAC experimental has more  bands. Thus  the  assignment of the  Raman characteristic peaks should be revisited. It will be probably very nice  to verify the Raman DFT based on the comparison with the NIC , SAC and CNZ  only.

Response 1: Thanks for the reviewer’s suggestion. As the reviewer pointed out, the PBC module can be used to study periodical structure by periodic boundary, but the demand on computer configuration (such as memory and hard disk) is higher. However, our calculational conditions is too poor to support this module. As the reviewer mentioned, the experimental and calcultional results are not perfect match, it is because the calculation modes used in this manuscript were based on the single-molecule structure and inter-molecular forces within solid-state crystalline unit cell were ignored. And the more accurate structural models need to optimize by using a server with better performance, and it needs further study in the future work.

Point 2: Minor L64, L73 Cm-1 - -1 must be superscript etc …

Response 2: Thanks for the reviewer’s suggestion. The format of wavenumbers has been revised in the manuscript.

Point 3: Figure 3 For comparison  figure 3 must include not just the PXRD of the  co-crystals but also  the  ones  generated  from the CCDC single crystal structure.  The  PXRD of the starting  APIs may be provided as Supporting  information. Something similar has been provided for THz ( figure 4).

Response 3: Thanks for the reviewer’s suggestion. The PXRD of the starting APIs has been added in the Figure 3, and the corresponding content has been revised in the manuscript.

There are no conclusions but Discussion?

Response 4: Thanks for the reviewer’s suggestion. The titles of chapter 3 and 4 have been modified to “Results and Discussions” and “Conclusions” in the manuscript.

Reviewer 2 Report

Overall, the manuscript is interesting and shows a novel point in cocrystal characterisation. The low-wave Raman spectroscopy and THz absorption spectroscopy have shown to be a good tool to characterize this type of materials. However, there are few comments that should be addressed before publication: 

  1. Figures can be condensed. For example, fig 3 and some of the Tables. 
  2. Have the crystal structure of both cocrystal uploaded into Cambridge Database?
  3. Two points will be really worthy of investigation:  1) Can it be the cocrystal formation be track in situ for example during the slow evaporation process? 2) Can you use these data for chemometric analysis? Do these techniques have enough resolution to provide mutilinear regression models?
  4. The discussion is extremely short, the paper would benefit from a broader discussion. For example, could you identify different cocyrstal habits? or can you couple these technique to industrila manudacturing of cocrystals to asses formation? For exmaple during sprya coating.

Author Response

Overall, the manuscript is interesting and shows a novel point in cocrystal characterisation. The low-wave Raman spectroscopy and THz absorption spectroscopy have shown to be a good tool to characterize this type of materials. However, there are few comments that should be addressed before publication: 

Point 1: Figures can be condensed. For example, fig 3 and some of the Tables. 

Response 1: Thanks for the reviewer’s suggestion. All the figures and tables have been modified in the manuscript.

Point 2: Have the crystal structure of both cocrystal uploaded into Cambridge Database?

Response 2: Thanks for the reviewer’s suggestion. Both cocrystals have not been uploaded into Cambridge Database so far.

Point 3: Two points will be really worthy of investigation:  1) Can it be the cocrystal formation be track in situ for example during the slow evaporation process? 2) Can you use these data for chemometric analysis? Do these techniques have enough resolution to provide mutilinear regression models?  

Response 3: Thanks for the reviewer’s suggestion. All the experimental results of the two cocrystals are obtained from that have been prepared. However, the low-wavenumber Raman spectroscopy can be tracked in situ due to it label-free, nondestructive characteristics, as the reviewer mentioned. THz and low-wavenumber Raman spectroscopy can realize the classification and prediction, but this manuscript aims to investigate the low-frequency vibrational characteristics of CBZ cocrystals. In addition, there have been many researches on the mutiliear regresstion models through THz spectroscopy, but there is a little about low-wavenumber Raman spectroscopy technique. The resolution of the low-wavenumber Raman system is 2.5cm-1, which have enough resolution to provide mutilinear regresstion models, and it needs further investigate in future.

Point 4: The discussion is extremely short, the paper would benefit from a broader discussion. For example, could you identify different cocyrstal habits? or can you couple these technique to industrila manudacturing of cocrystals to asses formation? For exmaple during sprya coating.

Response 4: Thanks for the reviewer’s suggestion. THz and low-wavenumber spectroscopy technique can be applied in many fields, such as pharmaceutical, chemical field. The two techniques are sensitive to the collective vibration, environment information and structure information of crystal lattice, thus they can identify the different cocrystal habits which relates to the environment information. In addition, the low-wavenumber Raman spectroscopy can be used in the real-time reaction monitoring of crystal form, phase, or structural formations during formulation of chemicals and polymers due to its label-free and nondestructive. These contents have been appended in the revision.